# Delivery of Active Peptides by Self-Healing, Biocompatible and Supramolecular Hydrogels

**DOI:** 10.3390/molecules28062528

**Published:** 2023-03-10

**Authors:** Seyedeh Rojin Shariati Pour, Sara Oddis, Marianna Barbalinardo, Paolo Ravarino, Massimiliano Cavallini, Jessica Fiori, Demetra Giuri, Claudia Tomasini

**Affiliations:** 1Department of Chemistry Giacomo Ciamician, University of Bologna, Via Selmi 2, 40126 Bologna, Italypaolo.ravarino2@unibo.it (P.R.); 2National Research Council, Institute for the Study of Nanostructured Materials (CNR-ISMN), Via P. Gobetti 101, 40129 Bologna, Italy

**Keywords:** Franz diffusion cells, hydrogels, low-molecular-weight gelators, peptides, self-assembly

## Abstract

Supramolecular and biocompatible hydrogels with a tunable pH ranging from 5.5 to 7.6 lead to a wide variety of formulations useful for many different topical applications compatible with the skin pH. An in vitro viability/cytotoxicity test of the gel components demonstrated that they are non-toxic, as the cells continue to proliferate after 48 h. An analysis of the mechanical properties demonstrates that the hydrogels have moderate strength and an excellent linear viscoelastic range with the absence of a proper breaking point, confirmed with thixotropy experiments. Two cosmetic active peptides (Trifluoroacetyl tripeptide-2 and Palmitoyl tripeptide-5) were successfully added to the hydrogels and their transdermal permeation was analysed with Franz diffusion cells. The liquid chromatography-mass spectrometry (HPLC-MS) analyses of the withdrawn samples from the receiving solutions showed that Trifluoroacetyl tripeptide-2 permeated in a considerable amount while almost no transdermal permeation of Palmitoyl tripeptide-5 was observed.

## 1. Introduction

Hydrogels are materials mainly composed of a 3D network of cross-linked fibers able to incorporate significant amounts of liquids, such as water or biological fluids. This is one of the reasons why they are among the most-used biomaterials in medical applications, not only for the production of healthcare products but also for the production of drug delivery systems, tissue engineering scaffolds and wound dressings [1,2]. A recent possibility to form hydrogels for these applications is focused on the use of small molecules instead of polymers [3,4]. The use of small molecules able to gelate water and organic solvents is an innovative process that is highly desirable for the formation of smart and sustainable materials.

These compounds, often called low-molecular-weight (LMW) gelators, are able to self-assemble through supramolecular interactions, such as hydrogen bonds, π–π stacking, hydrophobic interactions or electrostatic interactions, under given conditions induced by a trigger [5,6]. The trigger can be any energy input, physical or chemical, that reduces the solubility of the gelator molecules. Some of the most-used triggers to induce gel formation are pH [7] or temperature [8] changes, the addition of ions [9,10], light [11] and ultrasound sonication [12]. Supramolecular hydrogels are very versatile, because by tuning the gelator structure and the specific trigger to which it responds we can modify the final material properties and adapt them to the desired application. For example, their mechanical and morphological properties can be tuned depending on the tissue they should repair. Moreover, the non-covalent nature of their interactions makes them reversible materials, which can be disassembled with a specific stimulus and reformed again many times. Indeed, an important feature of these physical gels is thixotropy, i.e., the ability to reform after break, which also makes LMW hydrogels materials that can be injected and deliver drugs in situ [13,14].

Hydrogels based on amino acid derivatives present several advantages [5,15], such as easy and cost-effective synthesis and functionalization, structural diversity, low toxicity, high biocompatibility and biodegradability. Different gelators can also be mixed together to obtain a multicomponent gel, which may combine some of the properties of the two single components in the same material [16] or may be characterized by interesting new properties not accessible with a single component [17].

LMW gels can therefore be considered a valid alternative to polymers in many applications. In cosmetics, these compounds can be used for different purposes (rheological modifiers [18], skin care and topical therapies [19,20], perfume controlled releaser [21]), but have still received poor investigation.

In this work, we obtained a new multicomponent gel using two gelators. By mixing the two gelators in different ratios, we were able to tune both the mechanical properties and the pH of the final materials. The pH can be further modified with the addition of citric acid (currently used in many products applied on the skin [22,23]), thus obtaining a range of biocompatible hydrogels that can be used for topical delivery of drugs and for skin care applications. The pH of the skin is not an exact value but rather a range (4.1–7.4) and can widely vary depending on the anatomic site, age, gender and ethnicity [24]. Thus, having the possibility to finely tune the pH of the gel just by changing the ratio of three ingredients (two gelators and citric acid) can lead to a wide variety of formulations that can satisfy many different needs. To verify the permeation of the gel components, we performed in vitro tests using Franz diffusion cells. These represent a widely used methodology for studying transdermal delivery systems in vitro to measure substance permeation [25]. The gels were perfectly thixotropic; thus, they can be also used as injectable materials for drug delivery.

## 2. Results

We prepared supramolecular hydrogels formed by a mixture of the two peptide components Boc-L-Dopa(Bn)_2_OH **1** and Boc-L-Ala-Aib-L-Val-OH **2**. These materials are very cheap and may be obtained in multigram scale with a simple coupling reaction from commercially available amino acids (Figure 1). The details of the preparations and of the characterizations are reported in the Supporting Information (SI).

The gelation propensity of gelator **1** was previously reported by our group under several conditions [21,26,27] that never allowed us to obtain a hydrogel at neutral pH, needed for biological applications. The dissolution of gelators bearing a free acidic moiety at the *C*-terminus in water can occur in the presence of NaOH or other bases, usually at a pH of about 10. After the complete dissolution of the gelator, the gel formation can be triggered by the addition of several species, such as GdL (Glucono-δ-lactone), which protonates the carboxylates and lowers the pH, or CaCl_2_, whose Ca^2+^ ions act as physical crosslinkers between two carboxylates [7,9]. In our previous studies, we obtained gels with a pH of about 5 using GdL and about 8 using CaCl_2_, but we lacked a system that could provide the whole intermediate range of pH, from 5 to 8, just by tuning the quantity of trigger used. Such a system would be of great interest for applications in transdermal drug delivery, where gels and emulsions that can be applied topically and permeated through the skin can be exploited [4,28,29].

Gelator **2** has been previously studied by our group and is able to form gels in mixtures of water and organic solvents (H_2_O/EtOH, H_2_O/^i^PrOH) [30]. Despite our attempts using several triggers (GdL, calcium, arginine), this gelator is not able to form hydrogels but rather precipitates or crystalizes in the presence of water. Therefore, we used this tripeptide as trigger and co-gelator for the gelation of **1**.

Gelator **1** is not soluble in PB (phosphate buffer) solution, so it was dissolved in a basic NaOH solution (pH about 10). Gelator **2**, in contrast, is soluble in PB solution (pH 7.4, final concentration 0.03 M), and the resulting pH of the solution is about 6, depending on the amount of **2**. This second solution was added to the solution of gelator **1** and used to trigger the formation of the mixed gel. Different mixtures of **1** and **2** were combined to form gels with pH ranging between 5.5 and 7.4 (Table 1). All the hydrogels had an overall concentration of about 1 % w/V of the two components **1** and **2**. The role of **2** is to contribute to the gel formation and to reduce the overall pH, as it contains an acidic moiety. Hydrogels **A** and **B** contain an amount of **1** higher than **2** and have a pH > 7.0. As the introduction of a higher amount of **2** (hydrogel **C**) reduced the pH < 7.0, we reasoned that an additional amount of a biocompatible acid (citric acid) could further reduce the pH. In fact, in gel **D** the pH was < 6, suitable for skin applications, but a small precipitate formed right after the addition of citric acid and the gel did not form. Therefore, a different method for the gel preparation was developed. We heated separately the vials containing **1** and **2** (with citric acid) up to 60 °C and mixed them while hot (for details see Experimental section). Under these conditions, a self-supporting gel formed right after the hot mixing of the two solutions. So, in the following formulations this latter method was always used.

Hydrogels **C** and **E** were chosen for further characterizations as their pH is suitable for topical applications. The mechanical properties of hydrogels **C** and **E** were tested first by shaking the samples and then with the rheometer. Both samples showed excellent thixotropic properties in any case. In Figure 2, we show pictures of both samples after disruption by vigorous shaking and formation after rest of 16 h. The process may be repeated several times.

A more accurate analysis of the hydrogels’ rheological properties was obtained by measuring the amplitude sweep before and after breaking the gels (Figure 3a,b). Before break, both samples show a moderate strength of about 6 kPa and a long linear viscoelastic range (LVR). In particular, gel **C** shows a great elasticity and the absence of a proper breaking point (crossover between G’ and G’’). This behaviour was also confirmed by the step–strain experiment (Figure 3c) performed after breaking and reforming the samples two times: during the deformation step the G’’ modulus does not really overcome the G’, meaning that a proper breaking of the network did not occur with 100 % strain. The amplitude sweep experiment (Figure 3a,b) performed after break by vigorous shaking and reformation (16 h) showed that gel **C-R** has a slightly lower G’, while sample **E-R** completely recovers the previous stiffness. The step–strain experiments (Figure 3c,d) showed that both samples recover their G’ after repeated disruption of the network. This happens for the different stress applied to the material, which in the case of the shaking is higher than in the case of the rheological analysis.

The hydrogels obtained with this technique open the way to the formation of new materials that may be used for transdermal applications.

Measuring the amount of active substance that penetrates the skin is an important issue. However, it is not always possible to test the substance on skin. Therefore, different in vitro methods have been developed to measure the release. In vitro release studies can reveal critical information on the dosage and behaviour of the active substance as well as release kinetics. Franz diffusion cells are a widely used methodology in studying in vitro transdermal delivery systems to measure substance release and permeation.

In this work, the controlled permeation of some active peptides was analysed with the help of Franz diffusion cells. Pig ear skin was selected as the membrane [31]. As a first application, we checked the transdermal permeation of two tripeptides that have a strong cosmetic activity, as they act as anti-aging agents. TFA-L-Val-L-Tyr-L-Val **3** (trade name: Trifluoroacetyl tripeptide-2) is used in the commercial formulation Progeline^TM^ by Lucas Meyer Cosmetics for its unique mechanism of action on progerin synthesis modulation [32,33,34]. Pal-L-Lys-L-Val-L-Lys **4** (trade name: Palmitoyl tripeptide-5, also known as SYN^®^-COLL) has been widely used for its activity of collagen stimulation (Figure 4) [33,35].

The two molecules have been prepared in good quantities with high yields using liquid-phase synthesis, starting from commercially available amino acids (Appendix A). The introduction of the palmitic moiety in compound **4** was quite complicated due to the high lipophilicity of the fatty acid side chain. The details of the preparations and of the characterizations are reported in the Supporting Information. Following the same technique reported for the preparation of hydrogel **E**, we prepared the two hydrogels **F** and **G**, replacing the citric acid with the two active peptides **3** and **4** and mixing the two solutions at 40 °C for 2 min (Table 2), since the active ingredients can undergo degradation at high temperature. The two bioactive peptides were added to the formulation in a quantity of 1 mg/mL (0.1 % w/v), obtaining in both cases self-standing hydrogels with pH < 7.

The mechanical properties of hydrogels **F** and **G** were measured with thixotropy tests (hard shaking and recovery) and rheological analyses (amplitude sweeps and step–strain experiments).

These samples behave as the previously reported ones, as they both recover their solid structure after disruption on shaking the samples multiple times (Figure 5 and Figure 6). From the amplitude sweep analyses we can see that sample **F** has a lower G’ (3.9 KPa) and a higher elasticity compared to sample **G** (23.4 KPa). Even after the strong shaking, **F** does not convert to solution, but remains a viscous slurry, as can be seen in both Figure 5 (sample F-B) and Figure 6c, where G’ is always higher than G’’. This rheological behaviour means that the presence of **3** and **4** does not prevent gel formation, but the two active tripeptides slightly influence the rheological properties of the resulting gels. In particular, tripeptide **4** seems to positively interact with the gel network, as we can see from the increased G’. The ability of these gel networks to recover after multiple breaks paves the way to investigate the use of these gels as injectable tools for drug delivery [36,37,38].

The controlled permeation of the cosmetic active peptides **3** and **4** from hydrogels **F** and **G**, respectively, was analysed using diffusion in Franz cells, a widely used methodology to evaluate in vitro drug permeation [39]. These cells present some advantages, such as little handling of tissues, no continuous sample collection and a small amount of drug required for analysis. A membrane, synthetic or natural, separates the two compartments of Franz cells [40]. Synthetic membranes (polysulfone, cellulose or polydimethylsiloxane) must be inert, provide high permeability and not occlude the drug penetration [41,42]. Studies show that human skin is the best model in transdermal delivery systems, even though factors such as age, sex and race can affect characteristics of the tissue and these variabilities might change the permeation and diffusion through the membrane. However, the structural and biochemical characteristics of pig ear skin have been shown to provide comparable results to human skin. Therefore, in many studies, pig ear is used as the membrane in Franz cells [43].

In this specific study, aimed at a cosmetic application, the majority of the active peptides should be trapped in the skin layers, where their function is performed, and should not pass in the solution of the receiving chamber becoming systematically available [44,45,46]. The HPLC-MS analyses of the withdrawn samples from the receiving solutions (Figure 7) showed the percentage of the analytes permeated in the receiving chamber under the skin layer. After 24 h, 39 % of peptide **3** permeated through the skin (Figure 7A), while almost no permeation of peptide **4** was observed (Figure 7B), meaning that it is completely placed in the skin layer. Regarding the two gelators composing the formulation, after 24 h the permeation of Boc-L-Dopa(Bn)_2_OH **1** was 8 % and 6 % from hydrogels **F** (Figure 7A) and **G** (Figure 7B), respectively, while the permeation of Boc-L-Ala-Aib-L-Val-OH **2** was 46 % and 49 % from hydrogels **F** and **G**, respectively.

The viability/cytotoxicity of the individual components (**1**, **2**, **3** and **4**) of the gels was tested in vitro using a secondary human keratinocyte line. A quantity of 1 mg of each individual component was incubated with the cells for 24 and 48 h. The graph in Figure 8 shows that at both 24 and 48 h the individual components are non-toxic and the cells continue to proliferate after 48 h. Furthermore, in the viability graph we can see a slight increase in proliferative activity with the solution of **3** after 48 h.

## 3. Materials and Methods

**PB solution preparation.** The PB solution (0.1 M) was prepared by dissolving KH_2_PO_4_ in water and adjusting the pH to a final value of 7.4 by adding NaOH 1 M.

**Gel preparation (A-C).** Gelator **1** (quantity depending on the sample, see Table 3) was dissolved in a Sterilin vial by adding Milli-Q^®^ water and 1 eq. of NaOH and left under stirring for 2 h. Gelator **2** (quantity depending on the sample) was dissolved in a second vial using 0.670 mL of PB solution (0.1 M, pH 7.4). After the complete dissolution, the solution of vial 2 was added to vial 1. The final solution was gently swirled and then left to rest overnight (16 h).

**Gel preparation (D).** Gelator **1** (7 mg) was dissolved in a first Sterilin vial by adding Milli-Q^®^ water (1.275 mL) and 1 eq. of NaOH (0.015 mL) and left under stirring for 2 h. Gelator **2** (13 mg) was dissolved in a second vial using 0.670 mL of the PB solution (0.1 M, pH 7.4). After the complete dissolution of **2**, 0.040 mL of 1 M citric acid aqueous solution was added, and the formation of some precipitate was observed. The resulting solution was added to the first one, gently swirled for a few seconds and left to rest overnight (16 h). After this time, a viscous solution was obtained.

**Gel preparation (E).** The two vials were prepared with the same quantities and methods as gel **C** and then heated separately by placing them in a water bath (60 °C) for 5 min. This led to a complete dissolution of the precipitate in vial 2. At this point, vial 2 was added to vial 1 and left in the hot bath for few seconds while gently swirling, and then the vial was removed from the bath and left to cool down and rest overnight (16 h).

**Gel preparation (F).** Gelator **1** (7 mg) was dissolved in a first Sterilin vial by adding Milli-Q^®^ water (1.315 mL) and 1 eq. of 1 M NaOH (0.015 mL) and leaving under stirring for 2 h. Gelator **2** and the bioactive tripeptide **3** (2 mg) were dissolved in a second vial using the PB solution 0.1 M (0.670 mL, pH 7.4). After the complete dissolution, the two vials were heated separately by placing them in a water bath (40 °C) for 5 min. At this point, vial 2 was added in vial 1 and left in the hot bath for a few seconds while gently swirling, then the vial was removed from the bath and left to cool down and rest overnight (16 h).

**Gel preparation (G).** The same procedure of gel **F** was followed for the preparation of gel **G**, with the only difference that the bioactive tripeptide **4** was used instead of **3**.

**Rheological analysis.** All rheological analyses were performed using an Anton Paar (Graz, Austria) MCR102 Rheometer. A vane-and-cup measuring system was used, setting a gap of 2.1 mm. The gels were prepared as described and tested directly in the Sterilin Cup^®^, which fits in the rheometer. All analyses were performed at a fixed temperature of 23 °C controlled with an integrated Peltier system. Oscillatory amplitude sweep experiments (γ: 0.01–100 %) were performed using a constant angular frequency of 10 rad/s. Step–strain experiments were performed on gel samples that were already broken and reformed twice. The samples were subjected to consecutive deformation and recovery steps. The first step was performed by keeping the sample at a constant strain γ = 0.05%, i.e., within the LVE region, and at a fixed frequency of = 10 rad*s^−1^ for a period of 300 s, simulating the conditions of the gel at rest. The deformation step was performed by applying to the gel a constant strain of γ = 100 %, i.e., above the LVE region of the sample, for a period of 300 s. The recovery step was then performed at a constant strain γ = 0.05 %, i.e., within the LVE region, and at a fixed frequency of = 10 rad*s^−1^ for a period of 600 s. The deformation/recovery cycles were performed and repeated two times.

**In vitro membrane permeation test.** The membrane permeation test was conducted on a Copley Scientific Vertical Diffusion Cell HDT 1000 (Nottingham, NG4 2JY, United Kingdom) equipped with Copley Scientific Vertical Diffusion Cell (15 mm ×11 mL) Type “C” receptor. PBS (pH 7.4) was added to each receptor compartment, maintaining a temperature of 37 °C. The model membrane (pork ear skin, thickness: 1.2 ± 0.3 mm) was rinsed with physiological saline solution and then washed with PBS (pH 7.4). The membrane was clamped between the two compartments (receptor and donor). When the system was stable, 0.5 mL of gel formulation was applied to the skin membrane for each different diffusion cell. Samples (1 mL) were withdrawn from the sampling port of the receptor compartment at a regular time interval (0.5, 1, 1.5, 2, 4, 6 and 24 h) and analyzed using the HPLC-MS method described below. After each sample withdrawal, the receptor was refilled with an equal volume of PBS (pH 7.4) to maintain sink condition. For these studies, triplicate experiments were performed.

**Standard solutions and sample preparation for HPLC-MS analysis.** Stock standard solutions of Boc-L-Dopa(Bn)_2_OH **1**, Boc-L-Ala-Aib-L-Val-OH **2**, TFA-L-Val-L-Tyr-L-Val-OH and Pal-L-Lys-L-Val-L-Lys-OH were prepared by dissolving each compound at a concentration of 1.0 mg/mL in a mixture of 0.2 % formic acid in acetonitrile/0.2 % formic acid in water 30:70. Stock solutions were stored at 4 °C. Serial dilutions of stock solutions in the same mixture were prepared and calibration curves were plotted for all analytes. The linearity range was found to be between 0.01 and 20 µg/mL. All the analyses were performed in triplicate.

Sample preparation was performed by diluting the aliquots withdrawn from the sampling port of the receptor compartment of Franz cells, so that the analyte concentration would fall within the linearity range.

**HPLC-MS analysis.** HPLC-MS analyses were carried out on an Agilent 1260 Infinity II Chromatograph coupled with Mass Spectrometer MSD/XT equipped with electrospray ionization source and operating with a single quadrupole mass analyzer.

The ESI system employed a 5.0 V (positive polarity) spray voltage and a gas temperature 350 °C. The nebulizer gas and drying gas flows were 50 % and 12 L/min, respectively. The mass chromatograms were acquired in total ion current (TIC) modality from 50 to 3000 m/z and in single-ion monitoring (SIM) mode on the ESI-generated most abundant ion for each analyte; 396 m/z for Boc-L-Dopa(Bn)_2_OH, 306 m/z for Pal-L-Lys-L-Val-L-Lys or Palmitoyl tripeptide-5, 378 m/z for Boc-L-Ala-Aib-L-Val-OH, 498 m/z for TFA-L-Val-L-Tyr-L-Val or Trifluoroacetyl tripeptide-2. The chromatographic analyses were conducted on a Poroshell 120 EC-C18 (Agilent Technologies, USA) column (100 mm, 3.0 mm, 5 µm particle size). The mobile phase was composed of phase A (0.2 % formic acid in acetonitrile) and phase B (0.2 % formic acid in water). The linear gradient elution was A:B 30:70 (*v/v*) to A:B 90:10 (*v/v*) in 15 min at a flow rate of 0.4 mL/min. The re-equilibrium time was 3 min. The injection volume was 5 µL.

**Cell viability.** Molecules **2**, **3** and **4** were dissolved in DPBS 1x (purchased from Merck), while **1** was dissolved in NaOH 1M (1 eq.) because of its insolubility in the above-mentioned buffer, all at a concentration of 2 mg/mL. Volumes of 0.5 mL of each solution (1 mg) were deposited on the multiwell. The solutions of the individual components of the gels were added after reaching a confluence of 60 % within the well.

Human immortalized keratinocyte (HaCaT) cells were cultured under standard conditions in MEM medium supplemented with 10 % (*v/v*) FBS, 2 mM L-glutamine, 100 U mL^−1^ penicillin and 100 U mL^−1^ streptomycin. Cells were seeded on samples at a density of 10^5^ cells per cm^2^. The cells were incubated for 24 h and 28 h in a humidified incubator set at 37 °C.

The cell viability was determined with a resazurin reduction assay; the reagent is an oxidized form of the redox indicator that is blue in color and non-fluorescent. When incubated with viable cells, the reagent is reduced, and it changes its color from blue to red, becoming fluorescent. Briefly, cells were seeded on samples with complete medium. After incubation times, the resazurin reagent was added directly to the culture medium with a 10 % volume of medium contained in each sample and incubated for 4 h at 37 °C with 5 % CO_2_. Subsequently, aliquots from each sample were transferred to a 96 multiwell plate for fluorescence measurement at λ_exc_ 560 nm and λ_em_ 590 nm (Thermo Scientific Varioskan Flash Multimode Reader). We included a negative control of only medium without cells to determine the background signal and a positive control of 100 % reduced resazurin reagent without cells.

## 4. Conclusions

In this paper we reported a study on the preparation of biocompatible hydrogels with tunable pH, ranging from 5.5 to 7.6, that can lead to a wide variety of formulations useful for many different topical applications. The pH can be tuned either by changing the ratio between the two gelators or by adding citric acid.

All these hydrogels are thixotropic, and their mechanical properties were analysed with a rheometer before and after breaking the gels. In any case, both samples show a moderate strength and an excellent linear viscoelastic range with the absence of a proper breaking point in the studied range of strain, confirmed with thixotropy experiments.

Then, two cosmetic active peptides (Trifluoroacetyl tripeptide-2 and Palmitoyl tripeptide-5) were successfully added to the hydrogels without modifying their final pH and rheological properties and their controlled permeation was analysed with the help of Franz diffusion cells. Pig ear skin was selected as the membrane. The HPLC-MS analyses of the samples withdrawn from the receiving solutions showed that about 40% of peptide **3** passed below the skin layer while almost no permeation of peptide **4** was observed, meaning that it is completely absorbed in the skin layer. The viability/cytotoxicity of the individual components (**1**, **2**, **3** and **4**) of the gels was tested in vitro using a secondary human keratinocyte line, demonstrating that the individual components are non-toxic and the cells continue to proliferate after 48 h.

In conclusion, we demonstrated that these hydrogels are tuneable and biocompatible media suitable for the transdermal delivery of organic molecules.

## Figures and Tables

**Figure 1 molecules-28-02528-f001:**
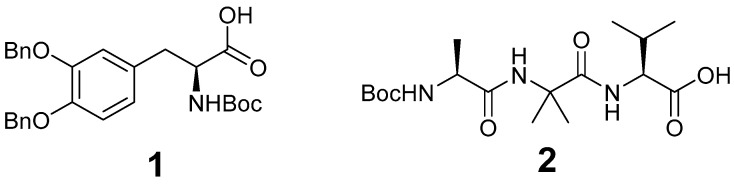
Chemical structure of the peptide gelators described in this work.

**Figure 2 molecules-28-02528-f002:**
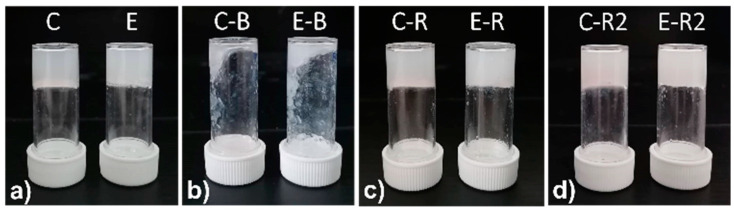
Thixotropy tests of samples **C** and **E**. From left to right: (**a**) samples after formation (**C**, **E**), (**b**) samples after break (**C-B**, **E-B**), (**c**) samples after overnight recovery (**C-R**, **E-R**), (**d**) samples after being shaken and reformed a second time (**C-R2**, **E-R2**).

**Figure 3 molecules-28-02528-f003:**
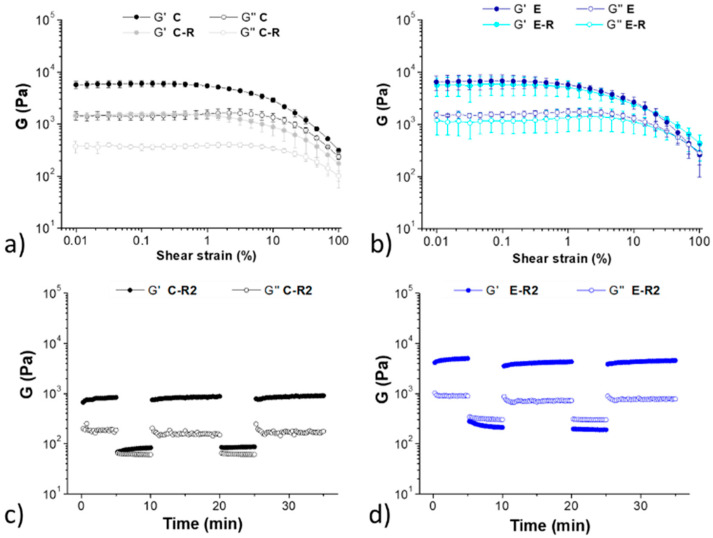
Top: amplitude sweep experiments of the hydrogel **C** (**a**) before (black) and after (gray) a break/recovery cycle. Amplitude sweep experiments of the hydrogel **E** (**b**) before (blue) and after (cyan) a break/recovery cycle. The analyses were repeated in triplicate. Bottom: step–strain experiments of hydrogels **C-R2** (**c**) and **E-R2** (**d**).

**Figure 4 molecules-28-02528-f004:**
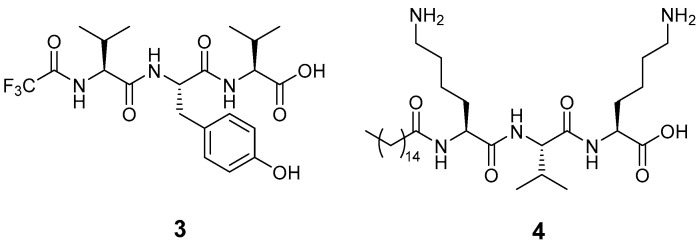
Chemical structure of the active peptides **3** and **4** described in this work.

**Figure 5 molecules-28-02528-f005:**
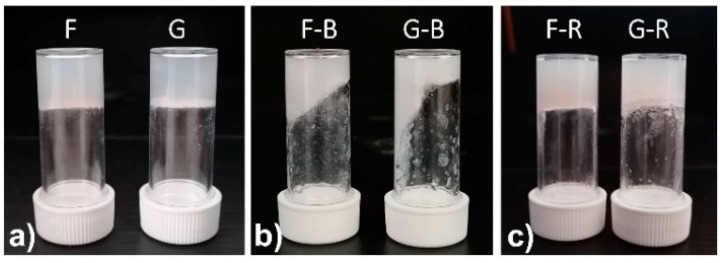
Thixotropy tests of samples **F** and **G**: (**a**) samples after formation (**F**, **G**); (**b**) samples after break (**F-B**, **G-B**); (**c**) samples after overnight recovery (**F-R**, **G-R**).

**Figure 6 molecules-28-02528-f006:**
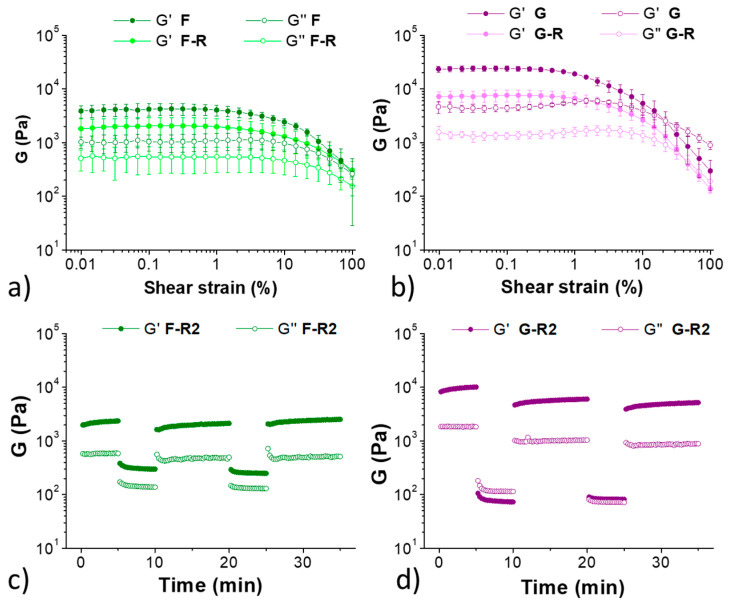
Top: amplitude sweep experiments of the hydrogel **F** (**a**) before (dark green) and after (light green) a break/recovery cycle. Amplitude sweep experiments of the hydrogel **G** (**b**) before (purple) and after (pink) a break/recovery cycle. The analyses were repeated in triplicate. Bottom: step–strain experiments of hydrogels **F-R2** (**c**) and **G-R2** (**d**).

**Figure 7 molecules-28-02528-f007:**
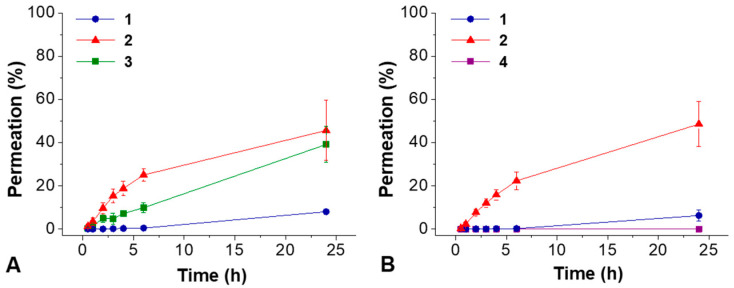
(**A**) hydrogel **F**, (**B**) hydrogel **G**. Percentages of Boc-L-Dopa(Bn)_2_OH (**1**)**,** Boc-L-Ala-Aib-L-Val-OH (**2**)**,** Trifluoroacetyl tripeptide-2 (**3**), Palmitoyl tripeptide-5 (**4**) permeated in 24 h, analyzed with HPLC-MS. These percentages represent the quantity of compounds passing below the skin layer.

**Figure 8 molecules-28-02528-f008:**
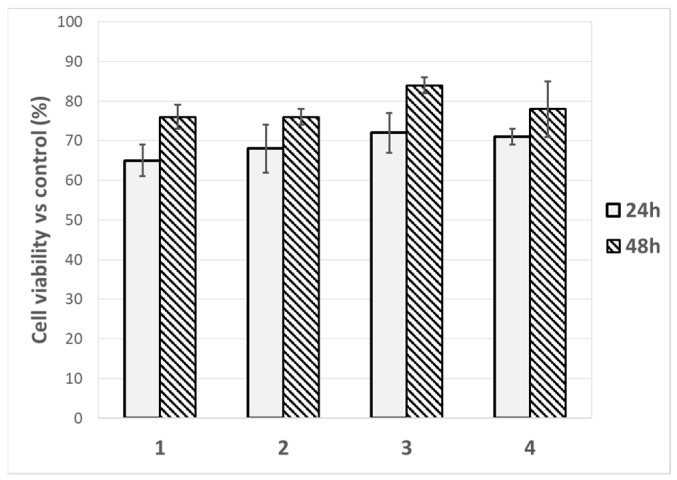
Cell viability of compounds **1**, **2**, **3** and **4** treated with HaCaT cells at 24 h and 48 h.

**Table 1 molecules-28-02528-t001:** Summary of the hydrogels containing peptides **1** and **2**. The % are expressed as w/V.

Gel	1	2	Citric Acid	T (°C)	Result	pH
**A**	0.65%	0.35%	-	23	Gel	7.60
**B**	0.65%	0.50%	-	23	Gel	7.30
**C**	0.35%	0.65%	-	23	Gel	6.80
**D**	0.35%	0.65%	0.38%	23	Sol	5.84
**E**	0.35%	0.65%	0.38%	60	Gel	5.55

Sol = solution.

**Table 2 molecules-28-02528-t002:** Summary of the hydrogels containing peptides **1** and **2** and the active peptides **3** (gel **F**) and **4** (gel **G**). The % are expressed as w/V.

Gel	1	2	Active Peptide	T (°C)	Result	pH
**F**	0.35%	0.65%	**3** (0.1%)	40	Gel	6.70
**G**	0.35%	0.65%	**4** (0.1%)	40	Gel	6.72

**Table 3 molecules-28-02528-t003:** Quantities used in the preparation of gels **A**–**C** prepared with different amounts of gelators **1** (Boc-L-Dopa(Bn)_2_OH) and **2** (Boc-L-Ala-Aib-L-Val-OH).

Gel	1 (mg)	1 M NaOH (µL)	H_2_O (mL)	2 (mg)	PB 0.1 M (mL)	Tot V (mL)
A	13	27	1.303	7	0.670	2
B	13	27	1.303	10	0.670	2
C	7	15	1.315	13	0.670	2

## Data Availability

The authors confirm that the data supporting the findings of this study are available within the article and its Appendix A.

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
