# Peer review of "Delivery of Active Peptides by Self-Healing, Biocompatible and Supramolecular Hydrogels"

_molecules, 2023, doi:10.3390/molecules28062528_

Round 1

Reviewer 1 Report

The author tried to report two small molecule peptide gel compounds, as well as their gel behavior and rheological properties when doped with citric acid and cosmetic active peptides, and investigated the transdermal release performance of peptide gel containing polypeptide 3 and polypeptide 4.
In general, the manuscript is written in a standardized way, with comprehensive data and reasonable discussion, so I recommend that it should be published in a journal. However, the following small issues need to be solved first:

1. There are a few errors in the text that need to be corrected. For example, there is no table2 in the text. In Table 2, the amount of peptide 3 and peptide 4 contained in gel F and gel G is not clear to the readers

2 Figure 7 (A) and Figure 7 (B) are not quoted in the text, which makes readers confused and difficult to read the expressed figure.

3. In the supporting information Scheme S1 and S2, the intermediates are not marked, only the target structures are expressed in Arabic numerals, while in the subsequent synthesis steps, the intermediates and target molecules are expressed in English abbreviations, which also makes readers confused. Therefore, it is recommended to use the same name for the compounds to be described in Scheme and synthesis steps

Author Response

The answer is in the attached file

Reviewer 2 Report

The theme of the publication is interesting and focus on the difficult material like peptides.

It seem to me that the authors do not distinguish between the release of the active substance and the penetration through the skin. The Franz diffusion cell were used properly, but the interpretation of the results is a bit confusing. For the release study for hydrogels usually synthetic membranes are used. When skin is used as the membrane we speak about permeation.  

I found some mistakes or inaccuracies in the text, I listed them below:

1.      Numbers of the tables have to be changed, name of table 1 was used twice for 2 different tables.

2.      In table on the page 5 (second table in the publication) description under the table that G- is gel could be confusing, because the sample name is G and the description of the result is also G. In my opinion the name of the sample should be changed.

3.      In table on page 5- composition of samples F and G is the same. What is the difference in the ingredients that cause different pH of received samples?

Also, the description in the 4th column: 3/4   is not clear which peptide was used or what is the ratio between them? Was them used together or it was 3 or 4 for two different samples called F and G?

4.      In table 3 ingredients used: the description what is 1 and 2 would be clear for the reader.

5.      On the fig. 7, in my opinion on the y axis the name should be: permeation.

When the skin is used in the study the permeation is investigated. When synthetic membranes are used in the study it should be called release.

Also line 236: “After 24 h, peptide 3 in 39 %, while almost no release of peptide 4 was observed, this meaning that it is completely placed in the skin layer.”

In my opinion It should be: “After 24 h, peptide 3 in 39 %, while almost no permeation of peptide 4 was observed, this meaning that it is completely placed in the skin layer.”

And further in all the section with the description of the study on diffusion Franz cell when skin was used there should be permeation used.

In the name of fig. 7 also permeation should be used not release.

6.      The description of preparation of gels F-G shows (line 287) that this two samples are the same. Maybe there was missed something in the description?

Author Response

The answer is in the attached file.
